# A Comprehensive Survey of Knowledge Graph-Based Recommender Systems: Technologies, Development, and Contributions

Janneth Chicaiza * and Priscila Valdiviezo-Diaz

Department of Computer Science, Universidad Técnica Particular de Loja, Loja 110105, Ecuador; pmvaldiviezo@utpl.edu.ec
* Correspondence: jachicaiza@utpl.edu.ec

**Abstract:** In recent years, the use of recommender systems has become popular on the web. To improve recommendation performance, usage, and scalability, the research has evolved by producing several generations of recommender systems. There is much literature about it, although most proposals focus on traditional methods' theories and applications. Recently, knowledge graph-based recommendations have attracted attention in academia and the industry because they can alleviate information sparsity and performance problems. We found only two studies that analyze the recommendation system's role over graphs, but they focus on specific recommendation methods. This survey attempts to cover a broader analysis from a set of selected papers. In summary, the contributions of this paper are as follows: (1) we explore traditional and more recent developments of filtering methods for a recommender system, (2) we identify and analyze proposals related to knowledge graph-based recommender systems, (3) we present the most relevant contributions using an application domain, and (4) we outline future directions of research in the domain of recommender systems. As the main survey result, we found that the use of knowledge graphs for recommendations is an efficient way to leverage and connect a user's and an item's knowledge, thus providing more precise results for users.

**Keywords:** knowledge graph; recommendation; survey; technologies; application domain

## 1. Introduction

In recent years, the use of recommender systems (RS) has become popular on the web. From the user's perspective, this type of system helps alleviate information overload because users receive personalized content or resources according to their profile or preferences. On the other hand, product or web content providers are betting on this type of service because they can capture the interest of their customers or users and, as a result, improve their sales or increase the use of the content they offer. For researchers, behind a recommender system [1], there is knowledge, processes, techniques, and challenges to address in order to improve the results and users' experience. In recent years, the emergence of knowledge graphs (KGs) has directed research on recommender systems towards the processing of this type of content. For a better understanding of the work that has been published so far on the subject, in this study, we identify and analyze the literature in which KG-based recommendation systems have been addressed.

Since their inception, recommender systems have been categorized as content-based (CB), collaborative filtering-based (CF), and hybrid (commonly content + collaborative + demographic) [2]. Currently, research related to recommender systems has increased considerably, creating new recommendation methods and combining different algorithms, ranging from more traditional [3] to hybrid [4] and knowledge graph-based methods [5].

Information filtering systems can be a key to discovering knowledge in an information-rich environment such as the web. However, traditional approaches do not have sufficient

capacity to reflect and exploit this knowledge [6]. A recommender system can leverage knowledge to build a semantic representation and to identify the most important entities and items for system users.

Today, KGs have become important resources to support tasks such as web searches, recommender systems, and question-answering systems. A knowledge graph is a structure that describes entities or concepts and connects them using different types of semantic relationships [6]. Graph-based recommendation methods are attracting attention in both academia and industry. In this study, we analyze 38 papers addressing the topics of recommendation and KG.

In domains such as education, RSs have become an important field of research [7]. Specifically, KG-based RS focus on supporting learners by finding educational resources [8] or relevant academic content that they can be integrated into their learning process [5,9,10]. Likewise, in other areas such as tourism, RS can deliver personalized information to users, thus enhancing their tourism experience [11–16]. Other domains upon which research on knowledge graphs has been focusing are health [17,18], entertainment [19], and business [14,20].

In addition, during the research, we came across studies that discuss knowledge graph applications in different domains [6,21–24], and their advantages and disadvantages, but only some are oriented on the analysis of the technologies used, their development, and their method of evaluation.

Regarding the state of research on KG-based RS, we found two studies where the topic is addressed: Liu et al. in [6] presented a survey focused on Knowledge Graph Embedding (KGE)-based recommendation methods, and Wang et al. [25] presented a tutorial focused on the recommendation problem from the perspective of graph learning and reasoning. When we broadened the search, we found multiple literature reviews or studies in the area of recommender systems, although most of them focused on theories and applications of traditional recommendation approaches.

Therefore, although there is a variety of literature on this subject, only two studies analyzed the role of KG-based RS [6,25]. The limitation is that they focus on a segment or subgroup of recommendation methods. In this paper, to overcome this information gap, we present the following contributions:

- We explore traditional and more recent developments of filtering methods for a recommender system.
- We identify and analyze proposals related to knowledge graph-based recommender systems.
- We present the most relevant contributions by application domain.
- We outline future directions of research in the domain of recommender systems.

The rest of the paper is organized as follows: Section 2 presents filtering approaches for recommender systems. Section 3 describes recommender systems based on knowledge graphs. Section 4 describes the methodology carried out in this survey. Section 5 shows the results of the literature review. Section 6 provides an overview of the future research directions. Finally, Section 7 presents the conclusions and future work.

## 2. Filtering Approaches for Recommendation Systems

Over the years of developing theory on recommender systems, three main generations of recommender systems have been developed. The first-generation RS (1995–2005) is based on three main approaches: content filtering, collaborative filtering, and hybrid methods; their methods are statistical or based on machine learning techniques. Second-generation RS (2003–2014) is based on context such as time, location, features such as ratings of user's group, etc. Research on this generation of RS is still ongoing, but third-generation RS is growing to be increasingly interesting. These RSs focus on semantic models of representation and the use of all knowledge components involved in the process of making recommendations [26].

### 2.1. Collaborative Filtering

The collaborative filtering approach (CF) is a traditional recommendation method that makes recommendations based on common user preferences and historical interactions [21]. This approach can be divided into memory-based methods and model-based methods. Memory-based methods, in turn, can be of two types: user-based and item-based. The most popular algorithm of memory-based approach is the KNN algorithm; this algorithm uses some traditional similarity measures such as Pearson correlation, Spearman, Coseno, Jaccard, etc. [2]. On the other hand, within the model-based methods, the most used are the factorization matrix (MF [27]) and its variants (NMF [28], SVD [29]).

Currently, new model-based collaborative filtering methods have been developed, for example Bayesian [30–32], clustering-based [33,34], rule-based [35], and graph-based [36] methods.

Collaborative filtering suffers mainly from two problems: sparsity of users' data when there are few interactions between the user and the items, and cold-start problem (new user and new item). An important aspect to consider is that traditional recommendation technology does not leverage semantic information, keyword relationships, and hierarchical structure [37].

### 2.2. Content-Based Filtering

A content-based recommender system learns to recommend items that are similar in terms of content features to those that the user liked in the past [38], i.e., this approach uses items' information to recommend based on user profiles.

Content-based recommenders can be classified into case-based reasoning [39] and attribute-based technique [40]. The case-based reasoning technique recommends items highly correlated with items that the user liked in the past. In contrast, the attribute-based technique recommends items based on matching their attributes with the user's profile.

Content-based filtering (CBF) suffers from some limitations such as overspecialization, limited content analysis, serendipity, and new user problems [41].

Most content-based recommender systems use simple models such as keyword matching or Vector Space Model (VSM) with Term Frequency-Inverse Document Frequency (TF-IDF) weighting [38], topical modeling to extract the semantic structures hidden in the document-based dataset by assigning topical distributions to the documents [42].

### 2.3. Demographic Filtering

Recommendation systems based on demographic filtering are based on the fact that users with certain common personal attributes (gender, age, country, etc.) also have common preferences [43]. Based on this, these systems can generate recommendations by categorizing users according to demographic attributes. These approaches are especially useful when the amount of item information is limited.

An advantage of demographic filtering is that it does not require user ratings of items that are necessary for content-based and collaborative filtering approaches.

However, this type of filtering has some disadvantages [44]: (1) Collecting complete information for users is not practical due to the security and privacy issues involved. (2) Demographic filtering is mainly based on user preferences, which forces the system to recommend the same item to users of related demographic groups.

### 2.4. Context Aware-Based Filtering

Context Aware-based Recommender System (CARS) is the most popular approach for incorporating context information. This approach assumes that the context is defined with a predefined set of observable attributes, for which the structure does not change significantly over time [41]. Context information such as time, location, geometric information, or the accompaniment of other people (e.g., friends, girlfriend/boyfriend, relatives, or colleagues) has recently been considered in recommender systems [39,45,46].

Contextual information provides additional information for making recommendations, especially for applications where it is not sufficient to consider only users and items [47].

### 2.5. Knowledge-Based Filtering

A knowledge-based recommender system suggests items to the user based on domain knowledge about how the items satisfy the user's preferences [48]. According to [38], these systems should employ three types of knowledge: knowledge about the users, knowledge about the items, and knowledge about the correspondence between the item and the user's needs.

Knowledge graphs can provide complementary information to overcome the problems faced by collaborative and content-based filtering approaches [21], since their recommendations are not linked to ratings; instead, they use domain knowledge. However, the main drawback of knowledge-based recommenders is that their creation involves having skills in knowledge engineering [49].

On the other hand, the semantic relationships present in a KG can be used by the system to improve its accuracy and to increase the recommended items' diversity. Based on this feature, novel KG-based approaches have been developed on the basis of classical approaches. For example, Reference [50] presented an approach for collaborative filtering with implicit comments where interactions between users and items are learned using a knowledge graph embedding method. Another proposal was explained in [51]; in this case, the recommender system for E-commerce uses a knowledge base to identify the domain knowledge of users, items, and the relationships between them.

In keeping with the idea that knowledge-based approaches do not experience the problems of traditional methods, this survey focuses on analyzing studies and applications developed with knowledge-based technologies for the recommendation process.

### 2.6. Hybrid Filtering

These systems commonly combine collaborative filtering with content-based filtering or collaborative filtering with any other recommendation approach. The goal of combination is to leverage each approach's advantages and to improve the overall system performance [38].

Currently, some works on hybrid approaches are based on deep learning methods [52,53], Bayesian networks [54,55], clustering [56,57], latent features [58,59], and graphs [60,61].

## 3. Recommender Systems over Knowledge Graphs

In 2012, Google proposed the term knowledge graph to refer to the use of semantic knowledge in web searches. Although the term associated with KG is new, representing pieces of knowledge as interconnected nodes is not a new idea. We can consider current KGs as an evolution of semantic networks, an old concept that emerged from the literature on cognitive science and artificial intelligence.

When Google created its KG, the purpose was to improve the search engine's capability and to enhance users' search experience [21]. Today, several companies and researchers have joined this movement and have created different KGs to describe specific domains. Before the term "knowledge graph" became popular, DBPedia and other linked data (LO) sets were generated thanks to Semantic Web technologies and the Linked Data—Design Issues proposed by Berners-Lee.

A knowledge graph provides machine-readable data organized as a graph; graph-data describes and interconnects entities of an open or a close domain. The data in a KG is accessible through the web and can be consumed automatically; these characteristics have facilitated the creation of applications such as [21] question-answering, recommender systems, information retrieval, domain-specific applications by knowledge area, and other applications (social networks and geoscience).

Concretely, KG recommendation exploits the connections between entities representing the users, the items to be recommendeded, and their interactions. The connections,

explicit or not, are used by the system to identify items that may be interesting or useful to the target user [62]. Thus, relationships provide the KG-based recommender with additional valuable information to apply inference between nodes to discover new connections [63]. On the contrary, in general, classical recommendation methods based on feature vectors overlook such connections, which may result in suboptimal performance, especially when there is data sparsity [25].

The availability of knowledge graphs in different domains has motivated researchers to conduct studies on knowledge graph-based recommendation algorithms. According to [6], current KG-based recommendation approaches can be classified into three categories: ontologies-based recommendation, linked open data-based recommendation, and knowledge graph embeddings-based recommendation. Furthermore, according to [21], KG-based recommendation can be classified into path-based approaches.

### 3.1. Ontology-Based (OB) Recommendation

In this approach, ontologies are used to model knowledge about users and their context, knowledge about the items, and knowledge about the domain.

In addition, the structure and semantics defined by an ontology facilitate the creation of rules to generate recommendations based on explicitly specified constraint or rules. That is, elements that satisfy the rules with respect to a given set of user's requirements are generated as a recommendation [64].

As explained by [38], similar to knowledge-based recommender systems, ontology-based recommenders do not experience most of the problems associated with conventional recommender systems, i.e., cold-start, sparsity of rating data, and overspecialization problems. The above is possible to achieve due to the fact that recommendations are based on user, item, and domain knowledge rather than based on user ratings.

Although OB recommendation exhibits advantages as indicated above, there are also some disadvantages related to the creation of an ontology, since this process is time-consuming and depends on an expert. To overcome this challenge, there are some proposals that demonstrate that the ontology creation process is possible to automate [38].

### 3.2. Linked Open Data (LOD)-Based Recommendation

Rich semantic information queryable from LOD data sets can enrich the system information, thus finding similar attributes among items to be recommended. The underlying advantage of doing this is to overcome the problem of data sparsity. On the other hand, as the recommendation process depends on external data, then the integrity of the external data may affect the recommendation results [6].

### 3.3. Embedding-Based Recommendation

In recent years, Knowledge Graph Embedding (KGE) techniques are becoming increasingly popular because they offer a simple and efficient way to generate recommendations [6]. In this case, the KG is transformed by using KGE algorithms; then, a recommendation framework can leverage the learned entity and relationship embeddings to produce a set of results [21].

An embedded KG can represent entities and relationships in a continuous vector space while preserving certain network information. The goal of applying KGE methods is to simplify the processing of a KG while maintaining its structure. In recommendation, KGE can be used to enrich the information of users and items, then the embedding representations can be used to calculate the similarity between both [6]. Among the most popular KGE models are TransE, TransH, TransD, and TransR [65].

In general, the introduction of KGE in recommender systems consists of using traditional recommender algorithms. Liu et al. [6] stated that, according to the relationship between knowledge graph embedding and the recommendation algorithm, there are two main ways to realize these tasks: independently learning and jointly learning. In this study, the authors analyzed different recommendation proposals based on this type of approach.

*3.4. Path-Based Recommendation*

The path-based recommendation is a natural and intuitive way to use KGs in recommendation processes. The algorithms attempt to explore various patterns of connections between nodes in a KG to retrieve additional information for recommendations. This method relies on hand-crafted designed meta-paths, which are difficult to optimize in practice, and it is not possible to design in some particular scenarios where entities and their relationships are not within a specific domain [21].

## 4. Methodology

In this section, we describe the tasks performed to find the literature related to the application of recommendation methods based on knowledge graphs. Figure 1 shows the general flow of the executed process that consists of three stages: search, selection, and analysis. The results of the last stage are discussed in Section 5.

**Figure 1.** Methodology: stages and tasks carried on to search, select, and analyze the related literature.

*4.1. Search*

To find literature related to the topics of interest, we conducted a systematic method of searching based on keywords and used Scopus as a source of information. To determine the most relevant studies in the area, we used Scopus's advanced search interface to filter the results.

To identify the documents indexed by Scopus, we designed a search that combined: (1) keywords related to the subjects (KG and RS), (2) keywords related to types of studies (survey, mapping, review, etc.), and (3) publication period between the years 2014 and 2020.

Listing 1 defines the final search string that was run in Scopus and allowed us to retrieve 79 documents.

**Listing 1.** Search query.

```
ABS(content AND filtering )) AND (ABS("knowledge graph"))
AND
(TITLE-ABS(review) OR TITLE-ABS(survey) OR TITLE-ABS(state?of?the?art)
OR TITLE-ABS(SOTA) OR TITLE-ABS(mapping) OR TITLE(study))
AND
PUBYEAR > 2013  AND PUBYEAR < 2021
```

*4.2. Selection*

To identify relevant papers within the study area, we chose two characteristics of the articles as inclusion criteria: (a) document type = [Article | Proceeding Paper | Review], and (b) language = English. Applying these two filters, we discarded 33 papers: 30 because they were conference reviews and 3 because they were written in Chinese.

Based on the title and abstract of the 46 papers selected in the previous step, we performed a preliminary screening of papers to verify their inclusion or non-inclusion in the group of papers to be analyzed. In this second step, we established three reasons for exclusion from this study:

1. Relation with the topic of interest. If the document did not describe a study or did not refer to the application of recommender systems in knowledge graphs, then it was excluded.
2. Availability. If the document was not available online or if access to its complete content was not possible, then it was excluded.
3. Duplicity. If there was more than one paper on the same topic and corresponding to the same authors, the most recently published paper was chosen.

After applying the criteria above, the corpus of documents to be analyzed was finally reduced to 38 papers. Of this group of reference papers, 5% were reviews, 37% were journal articles, and 58% were conference papers.

Figure 2 presents the annual distribution of articles found in Scopus. Additionally, the figure presents two series, the one corresponding to the selected papers vs. the series of papers that were excluded because they did not meet the abovementioned criteria.

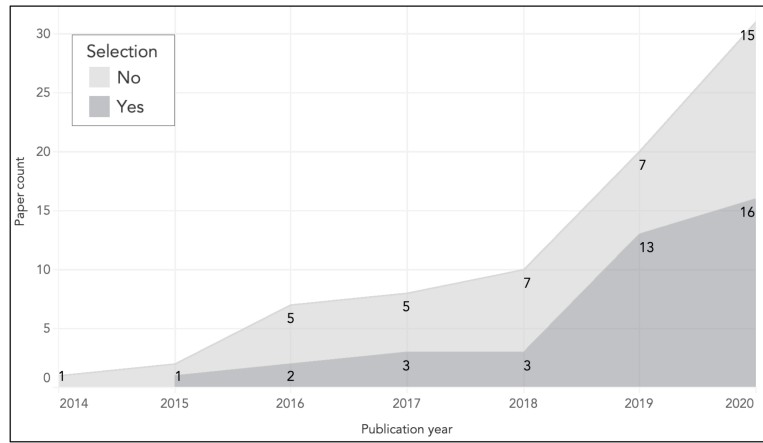

**Figure 2.** Annual production of papers indexed by Scopus. The stacked area plot shows the temporal evolution of papers related to the topic recommendation systems and knowledge graph.

As can be seen in Figure 2, in the last two years of the study, interest in the use of KG in recommender systems increased. Although scientific production grew in the area, a further increase is expected in the coming years considering the nature, heterogeneity, and large volume of data in the cloud.

To identify the most relevant terms in related works, we constructed term networks using the VOSviewer tool. Figure 3 presents the term network built based on the author

and index keywords, and Figure 4 presents the most frequent words found in the titles and abstracts of the papers.

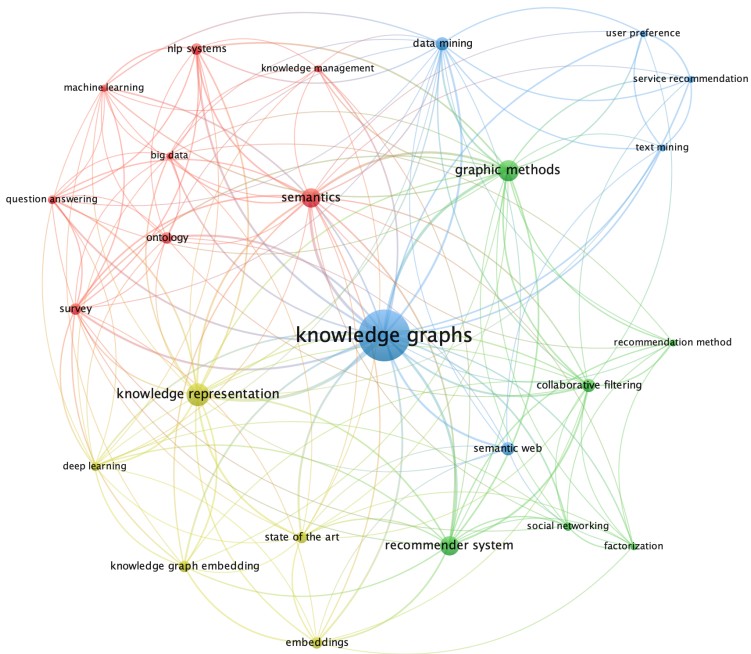

**Figure 3.** Network of more relevant keywords related to the selected papers.

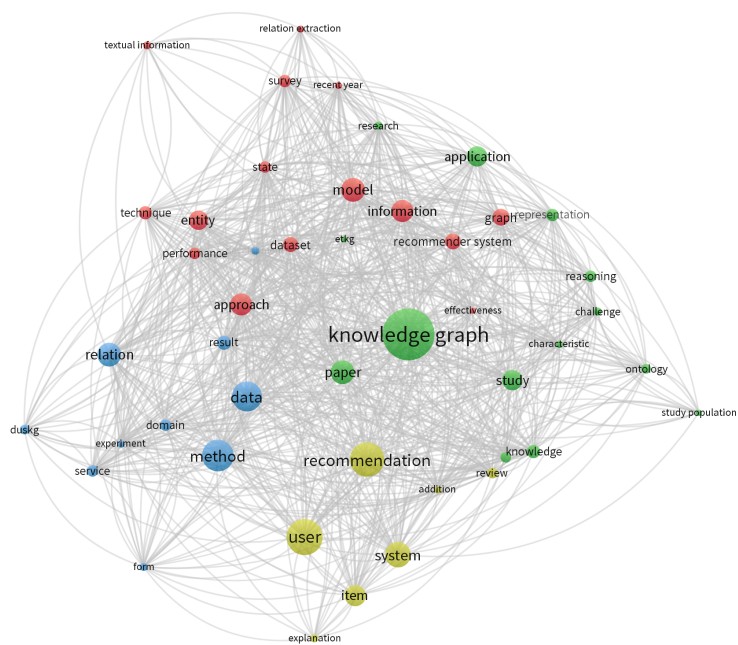

**Figure 4.** Terms network built from titles and abstracts of the selected papers.

As can be seen in Figures 3 and 4, some terms are present in both networks, such as knowledge graphs and recommender system. While other simple words such as data, user, item, or model only appear in the second network. In any case, the two figures highlight the terms discussed in this research.

*4.3. Analysis*

After we found the related papers to the topics of interest, we defined the categories and criteria used to organize and characterize them.

First, we identified the technologies and methods mentioned or used in each paper. Here, we considered two approaches: (a) technologies related to KG and semantic web, and (b) information recommendation or filtering methods. The results of this category of analysis are presented in Section 5.1.

Second, to determine the completeness or stage of maturity of each proposal, we classified each paper according to the following:

- Level of development. If the work presented a proposed high-level KG recommendation or if it referred to recommendations as a potential application of KGs, then the work was classified as conceptual. If the analyzed work presented a new recommendation method or if it presented the application of existing methods, then the work was classified as *implementation*.
- Type of evaluation. Regarding the type of evaluation carried out in each analyzed work, we considered three values: (1) experiments and comparative evaluation, if the authors performed experiments and compared their proposal with other baseline models, using performance metrics; (2) experimental results, if the proposal was evaluated using qualitative approaches or using exploratory experiments of the results; and (3) no evaluation, if no type of evaluation was found.

The analysis results are discussed in Section 5.2.

Finally, we attempt to organize the contribution of the selected papers according to the application domains. Here, we identify for each domain, the datasets used or mentioned in each paper, and the main findings reported in the literature. The results are available at Section 5.3.

## 5. Results

This section presents the results of the analysis carried out on 38 papers related to the topics of knowledge graphs and recommendation systems. In Section 4, we explained how we searched for and selected the analyzed literature.

The analysis results were organized into three categories of information. First, we identified the technologies or methods referred to, used, or proposed in each research. Second, we identify the level of development and type of evaluation carried out in the set of papers analyzed. Finally, we present the main contributions found by the application domain.

### *5.1. Technologies and Proposals*

In this first category of analysis, we classified the technologies into two subcategories: KG and Semantic Web Technologies, and Recommendation and Artificial Intelligence Methods.

### 5.1.1. Knowledge Graphs and Semantic Web Technologies

Knowledge Graph and Semantic Web offer an approach and a set of technologies for structuring, organizing, publishing, accessing, and processing linked data. Additionally, the organized data in a graph facilitates the construction of intelligent applications [5] such as recommender systems. Table 1 identifies the main technologies of this type that were found in the analyzed papers.

Table 1. Knowledge graphs and semantic technologies.

| Reference | KG and LD | KGE | Ontology | Query Language | Reasoning & Inference | Data Technology |
|---|---|---|---|---|---|---|
| [5] | | X | | | | |
| [6] | | X | | | | |
| [9] | | X | | | | |
| [10] | | X | | | | |
| [11] | X | | | | | |
| [13] | X | | | X | | |
| [14] | | X | | | | |
| [15] | X | | X | X | | X |
| [16] | X | | | | | |
| [17] | X | | X | | | X |
| [18] | X | | | | | |
| [19] | | X | | | | |
| [20] | X | X | | | X | |
| [21] | | X | | | | |
| [22] | X | | | | | |
| [23] | | X | | | | |
| [24] | | X | | | | |
| [25] | | X | | | | |
| [50] | X | X | | | | |
| [51] | X | | | | | |
| [63] | X | X | | | X | |
| [65] | | X | | | | |
| [66] | X | | X | | | X |
| [67] | X | | | | | X |
| [68] | | X | | | | |
| [69] | | X | | | X | |
| [70] | X | | X | | | X |
| [71] | | X | | | | |
| [72] | | X | | | | |
| [73] | X | | | | | |
| [74] | X | | X | | | |
| [75] | X | | X | | | |
| [76] | X | | X | | | |
| [77] | X | | X | | | |
| [78] | X | X | | | | |
| [79] | X | X | X | X | | |
| [80] | X | X | | | | |
| [81] | X | | X | | | X |
| Total | 23 | 21 | 10 | 3 | 3 | 6 |

As illustrated in Table 1, six groups of technologies or approaches are the most popular.

The most popular group corresponds to technologies related to KGs and linked data (LD). In some works such as [74,76,77], the authors bet on the reuse of linked datasets such as WordNet, DBPedia, Wikidata, Freebase, or Google Knowledge Graph. In other cases, the authors propose the creation of new KGs such as the Question-Answering KG [18], Event-centric Tourism Knowledge Graph (ETKG) [15], POI-Sensitive Knowledge Graph [20], Cohort Knowledge Graph [75], among others. Other works perform both tasks, i.e., create a KG (DBP-FB) and reuse other existing sets to enrich or complete information on the original graph [80]. Finally, another group of works corresponds to studies that discuss certain tasks related to KG [13,22,63,73,82].

Regarding the second group of technologies, more than 55% of the papers refer to KGE. Regarding the KG representation method (KGE), the use of a predominant method was not found. Different works highlight the use of different methods: TransE [10], TransD [68], RotatE [50], and TransH [20].

The third most popular group of technologies are ontologies. To support recommendation processes, among the ontologies created are Semantic Sensor Network Ontology [17],

Application Feedback Ontology [70], and Study Cohort Ontology (SCO) [66,75]. Some papers reuse ontological models available on the web, such as Semantic Science Integrated Ontology (SIO) [66], DBPedia ontology [77], and SKOS [79].

Other technologies that stand out in the other groups are SPARQL to query data from the KG [15,79], RDF to populate the KG with data [15,66,67], the Neo4J base to store graph data [70,81], among others.

### 5.1.2. Recommendation and Artificial Intelligence Methods

Table 2 shows the recommender approaches commonly used with KG technologies. One of the most commonly used approaches in combination with knowledge graphs is collaborative filtering. Methods such as neural networks, factorization matrix, deep learning methods, and Linear Support Vector are used to find user preferences. On the other hand, it can also be observed that KG-based methods use content-based filtering to generate recommendations based on the content of the items to be recommended. Most of the papers that incorporate this approach use similarity measures (e.g., Cosine) or Word2Vec to determine the similarity between different types of content or to cluster vectors of similar words.

Table 2 illustrates that some works use a knowledge-based approach as an alternative method to collaborative filtering in order to solve cold-start problems. These systems use reasoning techniques to exploit the information of the KG and inference methods to make recommendations.

The literature review results also show that another approach that can be used with KG is context-aware; however, few papers use this approach.

**Table 2.** Recommendation approaches used in the analyzed papers.

| Reference | Collaborative Filtering | Content Filtering | Context-Aware | Knowledge-Based |
|---|---|---|---|---|
| [5] | X | X | | X |
| [6] | | | | |
| [9] | | X | | |
| [10] | X | X | | |
| [11] | X | | | X |
| [13] | X | | | |
| [14] | X | X | | |
| [15] | X | | | X |
| [16] | X | | | X |
| [17] | | | X | |
| [18] | | X | | |
| [19] | X | X | | |
| [20] | X | | | X |
| [21] | | | | |
| [22] | | | | |
| [23] | | | | |
| [24] | X | x | | |
| [25] | X | | | |
| [50] | X | | | X |
| [51] | X | | | |
| [63] | | | | |
| [65] | | | | |
| [66] | | | | X |
| [67] | | | | |
| [68] | X | X | | |
| [69] | | | | |
| [70] | X | | | X |
| [71] | X | | | |
| [72] | X | | | X |
| [73] | | | | |
| [74] | X | X | | |
| [75] | | | | |
| [76] | X | X | | |

**Table 2.** *Cont.*

| Reference | Collaborative Filtering | Content Filtering | Context-Aware | Knowledge-Based |
|---|---|---|---|---|
| [77] | X | | | |
| [78] | X | | | |
| [79] | X | | | |
| [80] | | | | |
| [81] | X | | | |

As shown in Table 2, most of the analyzed works on KG employ a hybrid approach combining more than one recommendation method. Apart from these methods, there are other recommendation approaches that can also be combined with knowledge graphs, e.g., user group recommendation approaches [9,83], conversational approaches [84,85], and social network-based [86,87].

*5.2. Development and Evaluation Levels*

In this section, we summarize the level of development and type of evaluation carried out in the set of papers analyzed. Figure 5a indicates that approximately 6 out of 10 of the papers analyzed reached an implementation phase, either in the construction of the KG and/or the recommender system. Another set of these articles correspond to conceptual studies that do not contemplate the construction of the solution. Figure 5b demonstrates that a low percentage of the papers focus on experiments that do not contemplate comparison with other methods or algorithms. These proposals are mainly focused on evaluating user's satisfaction and opinion, usability and usefulness of the recommendations, effectiveness, efficiency, and robustness of the KG-based approaches. On the other hand, 18 papers did not include an evaluation process because they focused on surveys of techniques based on KG, recommender system architectures incorporating KG, identification of problems faced by knowledge graph methods, etc.

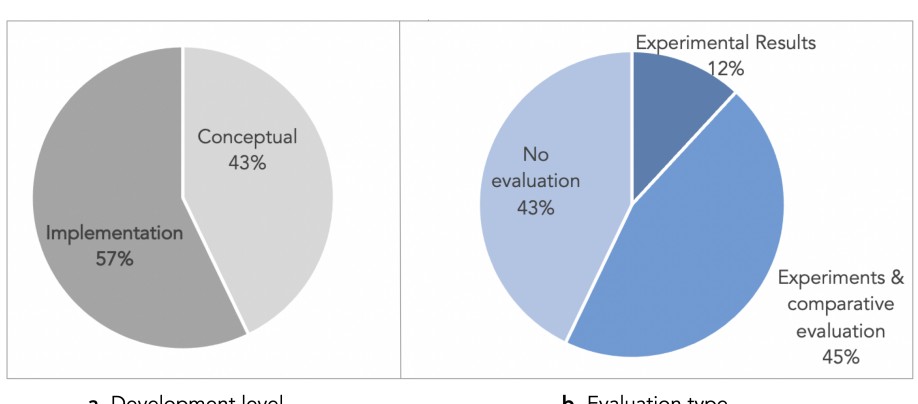

a. Development level                          b. Evaluation type

**Figure 5.** Development and evaluation type.

From Figure 5b, we can also conclude that a total of 24 papers include an experimental phase and a comparative evaluation with other methods. The results of these works are tested using some quality measures such as the following:

- Precision, recall, and F-measure are used when a recommender system must make decisions such as making or not making a recommendation. These measures are commonly used to evaluate the performance of recommender systems [88].
- Normalized Discounted Cumulative Gain (NDCG) allows for measuring the quality of the rank. NDCG is widely used to measure the effectiveness of recommendation algorithms [88].
- Mean Reciprocal Rank (MRR) measures whether the recommender system places the user's relevant items at the top of the list [3].

- Novelty and Diversity. Novelty measures the ability of RS to recommend items that appear novel to the user [89]. Conversely, diversity measures the ability of the RS to recommend items that are not similar to those preferred by the user in the past or that is not limited to recommending popular items only [90,91].

From the review conducted, we found that most of the papers that evaluate performance focus on measuring precision or NDCG while the other metrics are considered in less than 4% of the papers analyzed. This result indicates that current research is focused more on the evaluation of recommendation quality, entity-relationship ranking, and their relevance instead of evaluating, for example, the diversity and novelty of recommendations. Other measures that are little used are runtime, Mean Average Precision (MAP), and AUC-ROC.

### 5.3. Contributions by Application Domain

In this last category of analysis, we classified each paper according to the application domain, the recommended item, and the data source(s) used as the basis for generating the recommendations. Table 3 summarizes the results of this task.

**Table 3.** Data sources used by literature.

| Domain | Recommended Item | Datasource | References |
|---|---|---|---|
| Education | Labs | Course data | [9] |
| | Learning concepts | Educational resources | [5] |
| | Papers | ScienceDirect | [10] |
| Health | Questions | Online Suicide Rescue Instruction | [18] |
| | Treatments | Not specified | [66,75] |
| | User's behavior | CKAN | [17] |
| Lodging and Tourism | POIs | Hainan tourism | [15] |
| | | Social networks | [11] |
| | POIs, restaurant services & travel routes | Yelp dataset | [11,13,14] |
| | Hotels | Tripadvisor | [16] |
| Entertainment | Sound and music | Freesound.org | [74] |
| | | Last.fm | [50,74,78] |
| | | Songfacts.com | [74] |
| | | Facebook music | [76] |
| | Images | Pinterest | [50] |
| | | Facebook movies | [76] |
| | Movies | IMDB | [19] |
| | | Movielens | [50,71,78] |
| | | Not specified | [24] |
| | Books | Facebook books | [76] |
| | Not specified | Not specified | [69] |
| E-Commerce, business and Financial | Business services | Yelp dataset | [20] |
| | Insurance products | JGJISNF dataset | [68] |
| | Products | Retail e-Shop | [51] |
| | Toys | Amazon toys | [50] |
| | Books | Amazon books | [78] |
| | Music | Amazon music | [50] |
| | Financial actions | No specified | [79] |
| | Software apps | Apple App store | [70] |
| Cross-domain | Entities/relationships | Search logs | [72] |
| | | DBP15K, DWY100K, SRPRS | [80] |
| | | Wikipedia, Bengali documents | [77] |
| | Food industry | Amino Acid Dataset Freebase | [71] |
| | Language units | HowNet, Wikipedia & a newspaper | [81] |
| | Web resources | Not specified | [22] |
| | Not specified | Not specified | [6,21,25,63,65,67] |

As we can see in the Table 3, the most popular application domain for the KG-based recommendation is entertainment. For this domain, there are eight proposals out of the 38 analyzed. Additionally, here, we can observe that sound, music, and movie resources are the most recommended items. Moreover, some papers used more than one datasource to construct the KG and/or to carry out the recommendation experiments; for example, Reference [78] used two datasets to recommend music and movies.

The second most popular domain is E-commerce, business, and finances; in this case, the type of item recommended is diverse. The third group of applications is not focused on a particular area, i.e., proposals are cross-domain.

Another commercial area of application of RS is tourism and lodging; in this domain, the most popular dataset is Yelp, which has been used to generate different item recommendations such as POIs, restaurant services, and travel routes.

Finally, education and health are domains where recommendation systems are used to a lesser extent.

Below, we summarize the main paper's contributions for each application domain.

### 5.3.1. Education

As a contribution to the education sector, some KGs were created. In [9], a graph describing concepts and terminology of the cybersecurity domain was shown. Additionally, this KG contains information to help instructors and students track individual learning progress.

Interaction data sparsity during recommendation is an issue tackled in some proposals. The lack of rating data is a common scenario that can occur in academia, especially in open and non-formal education. To address this problem, Reference [10] proposed a hybrid model that recommends papers based on the browsing history of students of an academic search engine. Since the users of interest have no previous interactions in the recommender system, the proposed model can learn mappings between users' browsed papers and users' clicks on the recommended items.

Finally, in [5], an improvement in the accuracy of semantic similarity calculation between topics or concepts in a subject area is demonstrated. As the authors argue, the computation of semantic similarity plays a fundamental role in implementing a recommendation service for large educational datasets. According to the authors, current semantic similarity methods focus either on the structure of the semantic graph or only on the conceptual information. Therefore, in [5], a semantic similarity method is proposed, which combines these two methods and uses conceptual information to weigh the shortest path length between concepts. In the experiment, the authors demonstrate that their method has a certain degree of feasibility and credibility in calculating the semantic similarity of topics in KGs.

### 5.3.2. Health

In the health sector, we found four proposals related to knowledge graphs. In [66,75], the Study Cohort Ontology (SCO) is presented, which facilitates clinicians to perform population analysis and to generate cohort similarity visualizations. In this way, clinicians can select the most appropriate study populations for trial applicability.

Another study is [17], which deals with human care services [17]. In this case, data from different IoT sensors are used to recognize the current situation of a user and to predict future events. According to the authors, the current state of research about smart healthcare services focuses on analyzing user's behavior from single sensor data. Additionally, it focuses on analyzing and diagnosing the current situation of users. Therefore, a method is required to effectively manage and integrate a large amount of IoT sensor data. In the cloud, IoT sensor data stored in individual CKAN can be integrated based on common concepts. As a result, it is possible to generate an integrated knowledge graph considering data interoperability, and the underlying data are used as a base for prescriptive analysis.

Finally, in the healthcare field, the system proposed in [18] stands out. The system provides online instructions for volunteers of a suicide rescue organization called Tree Hole Rescue. The evaluation shows that the system provides answers, with high accuracy, to questions from users of the rescue organization. Additionally, the authors provide a set of methods to improve the system's question and to answer catalogue update capability.

### 5.3.3. Lodging and Tourism

Building KGs for lodging and tourism generally describe users and items (such as hotels, events, tourist attractions, and points of interest) as well as user–user or user–item interactions. Specifically, the KG proposed by [13] captures the logical associations between customer records (historical data on their movements and social communications).

Furthermore, Reference [11] showed that, by applying appropriate methods, the graphs can learn personalized weights of each user and the item to recommend, depending on the factors that most influence the customer's decision process. Another feature is that the graph can incorporate and connect heterogeneous information from location-based social networks (LBSN) into a unified representation space.

Another graph that can be highlighted is the Event-centric Tourism Knowledge Graph (ETKG), presented in [15]; the graph models the temporal and spatial dynamics of tourist trips. A relevant feature of ETKG is that it allows for representing dynamic data as activities that tourists can perform during a trip. The graph interconnects events using temporal relations in chronological order. Moreover, some additional features such as spatial information and attributes of journeys are incorporated into ETKG.

Regarding the recommendation, the proposed systems attempt to leverage networks of entities built from several sources. Concretely, the recommendation system proposed by [11] integrates relevant factors and features that influence user check-in behavior. In this study, the authors demonstrate that KG technologies allow for embedding heterogeneous information in unified representation space.

Moreover, KG-based recommendation can leverage weighted users' or items' properties according to their influence on the users' preferences. The method proposed by [16] can distinguish relevant customer's information to recommend hotels using a recurrent neural network and an attention-like mechanism.

In the tourism domain, we found that some works focus on addressing the sparsity of information and cold-start issues. Zhang, Wang, and Luo in [50] proposed the KGE-based collaborative filtering method that uses a deep neural network (DNN) to predict links that represent user–item interactions. Likewise, Reference [16] addresses a similar data completion task to predict user–item interactions. In [14], the authors tried to alleviate the cold-start problem by using the fine-grained restaurant services' features to identify similar users when a new user is coming.

Concerning the performance of the analyzed works, in general, an improvement is observed compared to other recommendation methods taken as a baseline. The RNN-based recommendation model of [16] predicts the interaction between users and hotels better (according to MAP and ROC-AUC) than the other three recommendation models. Likewise, the proposal of [11] is better in accuracy, recall, and MAP against eight different models, considering three evaluation datasets. Another method that improves the recommendation performance is ETKGCN [15], which recommends POIs related to tourist attractions. According to the authors, its performance is better because it integrates information from user reviews, a feature little exploited in other tourism networks.

In conclusion, the improvements achieved during recommendation are not only due to the use of appropriate recommendation or filtering methods but also due to the use of knowledge graphs.

### 5.3.4. Entertainment

In the entertainment domain, we found some works related to enrichment and link data with external sources. Regarding the enrichment of original data, Reference [69] pro-

poses to use data available in linked open data (LOD) datasets. Specifically, Reference [74] proposes to connect the data of sound and musical items with external graphs such as WordNet and DBPedia; thus, the initial data are enriched.

Regarding KG-based recommendation, from a practical point of view, we highlight here some particular features of some proposals focused on the recommendation of entertainment items:

- Dynamic data. Fischer et al. [72] proposed a system that recommends entities by taking advantage of the temporal nature of search log data. This approach can significantly improve the quality of recommendations compared to certain static models of relevance; particularly, it improves the freshness measure.
- Leveraging multiple variables and relationships. The recommendation proposal of [76] jointly exploits user ratings and item metadata. The recommendation engine of [74] combines different features such as semantic content and collaborative information obtained from implicit user feedback.
- Explainable recommendation. According to [19], ranking items and entities in the KG can serve as an explanation for recommendations.
- Reusing of existing technologies and models. The architecture of [72] is based on previous work. On the other hand, Reference [19] proposed pre-trained or built models that can be run without training, thereby allowing faster deployment in new domains.
- Performance improvement. Zhang, Wang, and Luo [50] demonstrated that the model KGECF achieves stable performance on five different datasets. In this case, among the reasons why KGECF is more stable is because (1) it can learn user's preference patterns more accurately and (2) the method used to embed the graph (RotatE) has very stable performance in modelling different types of relationships, such as one-to-many, many-to-one, and many-to-many relationships.

### 5.3.5. E-Commerce, Business, and Financial Sector

In e-commerce, certain items do not receive user feedback or do not have reviews, or there are users without historical data. To solve the lack of data, Reference [70] proposed an ontology as the driver to build users' profiles using multiple dimensions such as user's feedback, reviews, and user's ratings. In this way, the ontological model facilitates understanding the user-specific preferences by modelled from numerous perspectives.

Other works evidencing the use of multiple entities and data dimensions such as (1) Reference [68] present DCDIR, which utilizes a cross-domain mechanism to give personalized recommendations for new users in the insurance domain; (2) the KG built-in [20] incorporates data from supply-demand networks between business services and users, community network structures between users and between services, POIs, and detailed service content; (3) in [20], the KGE method called TransH is used to create dense representations of a KG; TransH facilitated the prediction of underlying relationships between users, POIs, and business services accurately.

On the other hand, for recommending business services, the experimental results of [20] demonstrate that the POIKG RS algorithm performs better than other collaborative filtering methods (Popularity, UserCF, ItemCF, and SPrank), especially when the data is sparse, which corresponds with a practical online scenario. Additionally, the approach can utilize the characteristics of different user groups and their POIs, resulting in better recommendation accuracy. Finally, POIKG RS presented significant online performance, i.e., when a user posts a comment on a service, they receive recommendation services immediately.

Unlike some proposals that focus on analyzing data sets from large companies such as Amazon, the system proposed by [51] was designed to be used by small-scale enterprise and retail shops, i.e., for companies where the number of users is small. As the authors emphasize, each user is important; therefore, the system should learn more about them based on their navigation through the website. To process KG based on user, product,

and activity data, a knowledge-based collaborative filtering technique is used in [51]. The efficiency of this approach (accuracy, recall, and NDCG) is better than that of the benchmark systems evaluated.

In the financial sector, in [79], the authors automatically personalize and contextualize user's actions to improve the effectiveness of customer contract risk management. The approach integrates business and external data into a knowledge graph and interprets profile-based actions through semantic reasoning over KG. One of the main contributions of the work is to help select actions that better mitigate risk related to financial actions. Unlike other proposals that use automatic metrics for evaluating the system performance, in [79], the authors use usability measures for business managers to assess their proposal.

5.3.6. Cross-Domain

In cross-domain, we place those proposals that do not explicitly define the scope of application. In this group, we find different types of research:

- Surveys that mention the recommendation term as (1) an application of KG [21,22], (2) KGE methods [65], (3) learning and reasoning on graph for recommendation (Wang2020c), (4) representation learning for dynamic graphs [63], (5) entity alignment [80], (6) relation extraction [73], (7) embedding mapping approaches [71], (8) knowledge base construction from unstructured text [67], and (9) extraction of semantic trees using KG [77].
- Survey on KG-based recommendation. In [6], KGE-based recommendation algorithms are presented. Additionally, the authors analyze the existing literature and compare five specific studies.
- Recommendation as secondary task. In [81], the authors present KG construction and the use of open source code to make recommendations. The graph is used to provide recommendations for language units.
- Recommendation as a central task. In [78], the model Knowledge-Aware Sequential Recommendation (KASR) is presented. KASR provides sequential recommendations, capturing both the sequence of interactive records and the semantic information in KG simultaneously. In this paper, the authors introduce the relation attention network to explicitly aggregate the high-order relevance in KG, and a unified knowledge-aware GRU directly plugs the significance into the modelling of interaction sequences. The authors have conducted experiments on three real-world datasets, and the results demonstrate that (1) knowledge-transfer based on relevant attributes helps to capture users' preferences more accurately and (2) the relation attention network mines the rich semantic information in KG.

## 6. Future Directions

There are some interesting directions to explore in future studies related to recommender systems based on knowledge graphs; however, in this section, we highlight four future directions that we find interesting to delve into.

### 6.1. Interpretability of Recommendations

Most recommendation models focus on achieving good performance from the point of view of accuracy. However, if the user cannot interpret the results, then the reliability of the system is reduced [3].

In KG-based recommender systems, the relationship between users and items can be easily interpreted from entities and relationships [16]. Furthermore, knowledge graphs contain rich semantic associations between entities, which can be used to strengthen the relationships between recommended items and to provide interpretability during recommendation [6].

In recent years, methods based on knowledge graphs have been proposed to interpret recommendations such as KPUP [92] and entity2rec [93], which in addition to presenting a higher accuracy, make recommendations easily interpretable. Although some works

have been developed using knowledge graphs, it is necessary to focus on how to use KG technologies to solve interpretability problems and to design interpretable models that lead to the explainability of the recommendation results.

### 6.2. Explainable Recommendation

Most of the research on explainable recommendations is based on unstructured data, such as text, images, audio, video stills, etc. However, if the recommender system possesses some knowledge about the recommendation domain, it facilitates the generation of personalized recommendations and explanations.

Currently, the advancement of KGE has made it possible to integrate graph embedding learning and recommendation techniques for improving the explanation of recommendations. Thus, the system can make recommendations with some domain knowledge and can tell the user why such items are recommended. For example, in [16], they develop a recommendation method to predict interactions between users and items using a KG and review text; in this case, explanations are generated based on the prediction made from the paths between a user and an item. Likewise, in [94], they use knowledge graphs to explain recommendations to users with items' unstructured textual description data.

Research on KG-based models for explainable recommendation represents one of the future directions for intelligent systems research since they can provide personalized recommendations in many research areas, such as personalized medical care, personalized online education, conversational systems, etc. Moreover, some studies on explainable recommendations [19,95] have demonstrated that explaining recommendations increase trust, transparency, and user acceptance in KG-based recommender system responses.

### 6.3. KG-Based Dynamic Recommendations

One of the few concerns of current KG-based recommendation algorithms is to make recommendations dynamic. Most KG-based recommender systems are considered static; although they present good accuracy results, very few consider users' dynamic preferences.

Dynamic recommendations are very important in areas such as healthcare where it is necessary to perform an evaluation of a patient's treatment over time [96], and in e-shopping [97] and social networks [98], where a user's interest can be influenced by other users very quickly.

Research that has focused on making recommendations dynamic is largely those based on traditional approaches such as collaborative filtering and/or content-based. Thus, a future direction is to leverage knowledge graphs for dynamic recommendations, for example, by capturing the attention of user's interests that change rapidly over time.

### 6.4. Learning with Knowledge Graphs

Representation learning has emerged to represent graphs. To create systems that can learn, reason, and generalize from this type of data, relational inductive biases need to be incorporated into deep learning architectures. These advances in graph representation learning have led to new state-of-the-art results in numerous domains such as recommender systems, question answering, and social network analysis [99].

Graph analysis (e.g., random walk) and graph learning (e.g., graph embedding) in recommendation models have achieved great success. Graph-based models have shown potential as the technologies for next-generation recommender systems. Recent efforts in graph representation learning such as KGAT [100], KGCN [101], and KGNN-LS [102] use GNN to synthesize information from such connectivity, strengthening the representation capability and enriching the relationships between a user and an item Wang2020c. These advances show the importance of exploring the potential of neural networks for KG-based recommender systems.

### 7. Conclusions

This survey presented a comprehensive review of studies related to KG-based recommender systems. Additionally, it identified the technologies used, the level of development, and the contributions of each related work. KG-based recommendation methods have huge potential applications in broad fields: education, healthcare, tourism, e-commerce, entertainment, etc. In this paper, we selected 38 papers, and in at least 8 of them [11,15,16,20,50,51,71,78], the authors found improved performance against baseline or traditional models. On the other hand, according to the work analyzed, there is a lack of complete proposals that include implementing and evaluating the proposals. In fact, 43% of the analyzed proposals have a reference or conceptual description of KG-based recommendation.

Studies on KG for recommender systems demonstrate that KGs are an efficient way to introduce user and item knowledge during the recommendation process. These studies are only the beginning of research in this area, and more research is needed on this topic. Future directions include interpretability and explainability, and the application of improved or hybrid methods by combining machine learning or deep learning techniques to improve the recommendation process. Based on the knowledge and overview obtained in this survey, as future work, the authors plan to design a KG-based recommender system for e-learning.

Finally, the future trends presented in this survey may inspire researchers to conduct further studies in the area of recommendation systems and KG.

**Author Contributions:** Introduction, J.C. and P.V.-D.; conceptual background, P.V.-D. and J.C.; methodology, J.C.; literature selection, J.C. and P.V.-D.; literature analysis, J.C. and P.V.-D.; research directions, P.V.-D. All authors have read and agreed to the published version of the manuscript.

**Funding:** This research was partially supported by the Universidad Técnica Particular de Loja and a scholarship provided by the Secretaría Nacional de Educación Superior, Ciencia y Tecnología of Ecuador (SENESCYT).

**Institutional Review Board Statement:** Not applicable.

**Informed Consent Statement:** Not applicable.

**Data Availability Statement:** Data sharing is not applicable to this review.

**Conflicts of Interest:** The authors declare no conflicts of interest.

### Abbreviations

The following abbreviations are used in this manuscript:

| | |
|---|---|
| CF | Collaborative Filtering |
| KG | Knowledge Graph |
| KGE | Knowledge Graph Embedding |
| LD | Linked Data |
| RS | Recommender System |

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
