# Peer review of "A Comprehensive Survey of Knowledge Graph-Based Recommender Systems: Technologies, Development, and Contributions"

_information, doi:10.3390/info12060232_

Round 1

Reviewer 1 Report

This is a bibliographic survey paper, but has invest a lot of work in classifying and presenting the Recommender Systems.

The paper presents a survey. Surveys are different from primary research. As such, it presents existing approaches. The main advantage of the paper is that it classifies and presents the surveyed aproaches in a very easy to understand way. The paper is well written and clear.

The paper is well structured and easy to read.

Author Response

Point 1: English language and style are fine/minor spell check required

Response 1: The document language has been reviewed, and we found some minor errors that were corrected.

Reviewer 2 Report

Dear Authors,

The proposed article makes the overview of the recommendation systems based on knowledge graphs, which is not new, but the authors cover 38 papers on this topic. It is somewhat confusing that in Table 2, there are 39 papers listed, so the number 38 may be wrong.

The authors showed that the most important domain for recommendation systems is Entertainment since most methods cover that topic. The second most important domain is E-commerce, Business, and Financial.

The authors then describe each of the domains and give some overview of the most important methods used there. Finally, they highlight four future directions of the future development of recommendation systems: Interpretability of Recommendations, Explainable Recommendation, KG-based Dynamic Recommendations, and Learning with Knowledge Graphs.

Unfortunately, the article is not structured well as the authors describe the methodology of their research after the results and predictions for the future are given. In addition to that, there are also some errors in the paper.

  • In Abstract, the authors use the abbreviation KG. Since the abbreviation is not defined yet, it would be better if the term “Knowledge Graph” would be used instead.
  • In line 256, the authors write: “Recommendation and Intelligence Artificial Methods.” I think the “Recommendation and Artificial Intelligence Methods” would be better.
  • The figure captions are short and give very little information about the figure content.
  • Table 2 has no caption at all.
  • The number of papers listed in Tables 1 and 2 is not the same. The authors should decide if they want to include 38 or 39 articles in their study.
  • Figure 1a has no reference in the text, so the authors should remove it since it is obviously not important for the paper.
  • In lines 283 and 284, the authors state: ”From Figure 1, part b, we can also conclude that the works that include an experimental phase and perform a comparative evaluation with other methods are based on some quality metrics such as:” this is not the case, without better explanation of the Figure 1b content.
  • In lines 297 and 298, the authors give interesting observation: “From the review conducted, we found that most of the papers that evaluate performance focus on measuring Precision or NDCG, while the other metrics are considered in less than 4% of the papers analyzed.” Did the authors try to find out what the reason for that is? The answer would add considerable value to their observation.
  • In line 428, the word “And” should be replaced with a more appropriate word for the beginning of the sentence.
  • In line 432, it is not clear to which method the word approach refers to.
  • In lines 545 – 553, the authors display some code (Expression 6.1 in text) without proper numbering and caption under the code.

Author Response

Point 1: The proposed article makes the overview of the recommendation systems based on knowledge graphs, which is not new, but the authors cover 38 papers on this topic. It is somewhat confusing that in Table 2, there are 39 papers listed, so the number 38 may be wrong.

Response 1: The number of selected papers was 38. In Table 2, a duplicate reference was found, and then it was removed.

Point 2: Unfortunately, the article is not structured well as the authors describe the methodology of their research after the results and predictions for the future are given. In addition to that, there are also some errors in the paper.

Response 2: In section 6, the survey methodology is explained as a process of 3 stages: search, selection, and analysis. And in section 4 (Results), we only described the results of the last stage. To explain better this link, we improve the writing of the first paragraphs of each section (4 and 6).

Point 3: In Abstract, the authors use the abbreviation KG. Since the abbreviation is not defined yet, it would be better if the term “Knowledge Graph” would be used instead.

Response 3: In the abstract, the abbreviation KG was changed by the Knowledge Graph term.

Point 4: In line 256, the authors write: “Recommendation and Intelligence Artificial Methods.” I think the “Recommendation and Artificial Intelligence Methods” would be better.

Response 4: The "Recommendation and Intelligence Artificial Methods" term was changed by "Recommendation and Artificial Intelligence Methods"

Point 5: The figure captions are short and give very little information about the figure content. Table 2 has no caption at all.

Response 5: Captions of figures 2, 3, 4 and 5 were updated with more details. In table 2, we put the caption "Recommendation approaches used in the analyzed papers".

Point 6: The number of papers listed in Tables 1 and 2 is not the same. The authors should decide if they want to include 38 or 39 articles in their study.

Response 6: The number of selected papers was 38. In Table 2, a duplicate reference was found, and then it was removed

Point 7: Figure 1a has no reference in the text, so the authors should remove it since it is obviously not important for the paper.

Response 7. Figure 1a is referenced in section 4.2 as Figure 1 (a).  

Point 8: In lines 283 and 284, the authors state: ”From Figure 1, part b, we can also conclude that the works that include an experimental phase and perform a comparative evaluation with other methods are based on some quality metrics such as:” this is not the case, without better explanation of the Figure 1b content.

Response 8: Figure 1(b) shows the evaluation type carried on each analyzed paper. We found three categories of evaluation. To explain better these categories, we add a new phrase describing the content of Figure 1(b). Also, we include the numbers of papers that are in each evaluation type.

Point 9: In lines 297 and 298, the authors give interesting observation: “From the review conducted, we found that most of the papers that evaluate performance focus on measuring Precision or NDCG, while the other metrics are considered in less than 4% of the papers analyzed.” Did the authors try to find out what the reason for that is? The answer would add considerable value to their observation.

Response 9: In lines 307 - 309, we explain the reason related to this finding.

Point 10: In line 428, the word “And” should be replaced with a more appropriate word for the beginning of the sentence.

Response 10: We updated the punctuation sign that separates each item of the enumerated list. Instead of ".", we used ";" and we removed the word "and".

Point 11: In line 432, it is not clear to which method the word approach refers to.

Response 11: In lines 443 and 444, we mention the algorithms that the POIKG RS algorithm was compared.

Point 12: In lines 545 – 553, the authors display some code (Expression 6.1 in text) without proper numbering and caption under the code.

Response 12. On page 17, we put the title for the expression (search query).

Reviewer 3 Report

This is a well-written survey of a currently-important area.  I find nothing in the paper to criticize.

A question that arises, reading the paper in a historical context, is the relationship between current KG systems and the semantic networks and other graph-based knowledge representations (e.g. Sowa's conceptual graphs) that were developed decades ago and then seemingly abandoned as ML technologies became dominant in applications.  Are current KG systems a rediscovery of these old techniques (with of course better implementation) or are they new in some fundamental way?

I do not require addressing this question for publication, only suggest it in case the authors are interested.

Author Response

Point 1: A question that arises, reading the paper in a historical context, is the relationship between current KG systems and the semantic networks and other graph-based knowledge representations (e.g. Sowa's conceptual graphs) that were developed decades ago and then seemingly abandoned as ML technologies became dominant in applications. Are current KG systems a rediscovery of these old techniques (with of course better implementation) or are they new in some fundamental way? I do not require addressing this question for publication, only suggest it in case the authors are interested.

Response 1: In 156-158, we introduce the semantic network term and its possible relationship with the KG term.

Round 2

Reviewer 2 Report

The corrections the authors did improved their article considerably but still the structure of the article remains inappropriate as the results of the research are presented prior the description of the used methods. Authors did respond to this consideration, but they kept the same structure as in original manuscript.  

Author Response

Point 1: Authors did respond to this consideration, but they kept the same structure as in original manuscript. 

Response 1: To improve the manuscript structure, we moved the Methodology section before the Results section and changed some references to them. For example, we updated the numeration of sections in the last paragraph of the introduction.